# Impact of Geometrical Imperfections on Estimation of Buckling and Limit Loads in a Silo Segment Using the Vibration Correlation Technique

**DOI:** 10.3390/ma14030567

**Published:** 2021-01-26

**Authors:** Łukasz Żmuda-Trzebiatowski, Piotr Iwicki

**Affiliations:** Faculty of Civil and Environmental Engineering, Gdansk University of Technology, Narutowicza Str. 11/12, 80-233 Gdańsk, Poland; luktrzeb@pg.edu.pl

**Keywords:** buckling, vibration correlation technique, thin-walled structures, steel structures, shell structures, non-destructive testing, finite element method

## Abstract

The paper examines effectiveness of the vibration correlation technique which allows determining the buckling or limit loads by means of measured natural frequencies of structures. A steel silo segment with a corrugated wall, stiffened with cold-formed channel section columns was analysed. The investigations included numerical analyses of: linear buckling, dynamic eigenvalue and geometrically static non-linear problems. Both perfect and imperfect geometries were considered. Initial geometrical imperfections included first and second buckling and vibration mode shapes with three amplitudes. The vibration correlation technique proved to be useful in estimating limit or buckling loads. It was very efficient in the case of small and medium imperfection magnitudes. The significant deviations between the predicted and calculated buckling and limit loads occurred when large imperfections were considered.

## 1. Introduction

Buckling is one of the most common reasons for steel structure failures. Thin-walled elements subjected to compression are very sensitive to stability loss, therefore it is essential to find a method to monitor the conditions of existing structures. For a long time, a common buckling test showed a major disadvantage—destructive nature. The tested loaded structures often buckled with plastic deformations, due to their imperfection sensitivity. Geometrical imperfections are hard to assess without professional scanning equipment. The numerical results of limit and buckling loads at a design stage can differ significantly from their real values. The destructive methods cannot be applied to examine the existing structures.

The first non-destructive methods employed for steel columns were the Southwell method [1] and the vibration correlation technique (VCT) [2]. The latter combines vibrations and stability of investigated structures since they show similarities between buckling and vibration behaviour (especially columns).

The VCT can be used either to determine actual boundary conditions or to predict the buckling loads [3].

In the case of axially loaded columns the relation between squared natural frequencies and applied load is linear [4]. This equation, in terms of relative variables reads:(ω_n_/ω_n0_)^2^ = 1 − P/P_n_,(1)
where: ω_n_ is the n-th natural frequency of the loaded structure, ω_n0_ is the n-th natural frequency of the unloaded structure, P is applied load and P_n_ is the buckling load corresponding to n-th vibration mode. When the n-th natural frequency equals zero, the compressive load is a corresponding n-th buckling load. Jubb et al. [5] stated that vibration and buckling modes need to be identical in shape in order for there to be a linear relationship between the squared natural structural frequency and the buckling load.

While measuring natural frequencies of an existing structure at two load stages it is possible to draw a line pointing the predicted buckling load. The only condition is the identity of buckling and vibration modes, thus preliminary numerical calculations to determine the first buckling mode and a sufficient number of applied sensors to specify the vibration modes are indispensable.

In the course of the VCT it is also possible to estimate real boundary conditions. It can be useful in validating numerical models of existing structures. Measuring natural frequencies of an erected structure and comparing them to the numerical values is the way to check the stiffness of supports in a numerical model. However, when the geometric imperfections are not measured, assessment of real boundary conditions by the VCT may be inaccurate for complex and imperfection-prone structures.

The relationship between the squared natural frequencies and the applied load is exactly linear only in the case of columns with simple supports [2], where the buckling and vibration modes are identical. Small deviations from linearity may be observed in compressed columns and plates or frame systems with different boundary conditions [6]. Nevertheless, the VCT can be still applied successfully. The problems with the VCT application occur when initial geometrical imperfections are taken into account, here considerable deviations from linearity are apparent [7].

Massonnet [6] tested uniform beams, plates and cylindrical shells with various supports and confirmed more or less linear relationship between squared natural frequencies and applied load of investigated elements.

The VCT has been tested on different structures such as thin rectangular plates [8,9], shells [10] or steel cylinders [11].

Franzoni et al. [12] presented various modifications to the standard relationship between vibrations and buckling proposed by researchers throughout the years.

Recent research has focused mainly on shells. Based on a series of experiments, Arbelo et al. [13] proposed a new modified VCT approach, plotting 1 − (f_n_/f_n0_)^2^ versus (1 − P/P_n_)^2^ using best-fit second order curve to find a squared drop of the load-carrying capacity ξ^2^ due to the initial imperfections. Next, estimation of the predicted buckling load (P_imp_) led to the following formula:P_imp_ = P_cr_ (1 − ξ).(2)

Skukis et al. [14] verified this method experimentally with a good result. Recent papers confirmed effectiveness of the modified VCT by conducting experimental and numerical analyses on cylindrical shells stiffened with lozenge grid cores [15], pressurised orthotropic shells with rectangular stiffeners [16], sandwich plates with iso-grid cores [17], composite unstiffened shells [18] and variable angle tow composite cylindrical shells [19].

Franzoni et al. [12] made an attempt to evaluate the relationship between the applied load and natural frequencies of a simple supported isotropic unstiffened cylindrical shell analytically. The analytical approach was successfully compared with numerical calculations.

The paper deals with problems of predicting the buckling loads of stiffened shell. The silo segment strengthened with cold-formed columns is investigated numerically, the VCT is applied to determine the buckling loads on the basis of calculated natural frequencies. Various geometrical imperfection shapes and amplitudes are introduced into the model. The VCT is applied to each numerical model. Distinct structural performance in relation to the literature-based pattern is observed here. The conducted calculations enhance the previous analysis of the same silo [20] by introducing geometrical imperfections and changing boundary conditions.

The research deals with part of a real structure, instead of a small experimental model. Usually investigation of the issue pertains to actual imperfections which were measured in experimental models. However, the specific imperfections to emerge in subsequent project stages are unknown. Generally, they are assumed as the first buckling mode or the first vibration mode, therefore the paper presents the impact of such imperfections on relation between squared natural frequencies and compressive force. In the introduced investigation it can be noticed that a zero frequency in the buckling case is not necessarily the first one, it may be a higher one as well. An innovative element of research concerns the impact of imperfection on limit loads and their determination using the VCT.

The VCT approach is an experimental procedure. The main advantage of the method is an accurate non-destructive prediction of the buckling load of a structure. However, before conducting the experiments theoretical or numerical analysis of the investigated problem should be performed. The main scope of the research was to perform numerical simulations of the imperfect silo segment in order to predict relationship between applied load and squared natural frequencies. As a result, the buckling or limit load was find both by means of the VCT and non-linear static analysis. It can be added that some experiments are difficult to perform due to the scale of the tested structures but the phenomenon may be investigated using numerical simulations.

## 2. Materials and Methods 

The investigated structure was a stiffened silo segment. Based on research [21,22], it can be concluded that the silo segment model describes well the behaviour of the entire silo in the case of sparsely distributed columns. The silo was 8.04 m in diameter and 17.62 m high. The wall included corrugated steel sheets with the 76 mm pitch, the 18 mm depth and the 0.75 mm thickness (Figure 1a). The silo was strengthened with 18 columns sparsely distributed along the circumference. The columns were made of cold-formed channel sections of 4 mm thickness and dimensions shown in Figure 1b. The entire structure material was steel S355 of the following parameters: specific weight 7850 kg/m^3^, elasticity modulus 210 GPa and Poisson’s ratio 0.3.

The investigated structure was modelled by a commercial ABAQUS package, version 6.14 [23]. The entire silo was substituted by the segment including the wall section of the 40 degrees angle and three columns (Figure 2a). One column was placed in the middle of the silo part and two column halves at the edges. The boundary conditions at the side edges were intended to simulate buckling of the entire silo [21]. A simplified model was introduced in order to limit the number of finite elements, consequently to shorten the computational time.

The lower edges of the silo segment were fixed, the upper edges restrained translations in radial and circumferential directions, the lateral edges of side columns constrained translations in a circumferential direction and rotations around radial and vertical axes. The columns were attached to the wall by means of point tie connectors simulating rivets or screws.

The silo wall, as well as the cold-formed columns, were modelled with 4-node shell elements with reduced integration. A simplified numerical model consisted of 200,340 finite elements. A single corrugation wave was approximated by 8 elements.

The load was imposed in the form of support displacement applied vertically to the nodes located on the upper edges of the columns. The silo wall was not loaded in order to avoid a number of local buckling modes of the corrugation sheets. A pattern of displacement load subjected to one of the columns is shown in Figure 3.

Both perfect and imperfect silo geometry cases were considered. Two primary free vibration (F1, F2) and buckling (B1, B2) mode shapes were implemented by the software inner procedure to the numerical model in the form of initial geometrical imperfections (Figure 4). Relevant linear buckling analysis was performed to copy the nodal coordinates of buckling modes and introducing such a new geometry in order to conduct further analyses. Imperfection amplitudes in each case were set as 2 mm (a), 5 mm (b) and 20 mm (c). The total number of the investigated structures was equal to 13.

Numerical calculations covered the following analyses for each silo geometry model:Linear buckling, resulting in 5 primary buckling modes and eigenvalues. The buckling modes were bound to correspond to the vibration modes, therefore it was essential to compare them in order to draw right conclusions. The buckling eigenvalues were used to assess the primary buckling loads.Geometrically non-linear analysis, leading to a full equilibrium path: the segment displacement load vs. the average vertical displacement of the columns top edges. A static, general method based on Full Newton solution technique was applied [23]. The initial, minimum and maximum increment sizes were equal to 0.001, 10^−15^ and 0.01, respectively. In order to avoid problems with solution convergence automatic stabilization was introduced in the models by numerical damping factor (equal to 10^−9^). The analytical aim was to determine the limit load, i.e., the load corresponding to the load-displacement path maximum (P_lim_).Combined geometrically non-linear and dynamic eigenvalue analysis. Non-linear analysis was performed repeatedly—each time the displacement load increased to a certain value in order to pre-stress the structure. After each process a dynamic eigenvalue problem was solved, resulting in 5 vibration modes and corresponding natural frequencies. The vibration modes were necessary to trace their shape variation during the loading course [20]. It was easy to confuse the natural frequencies of various orders as the pre-load increased. The primary natural frequencies allowed applying the vibration correlation technique and drawing the relationships between the applied load and the squared natural frequencies.

## 3. Results and Discussion

The aim of the performed analyses is to verify the vibration correlation technique in its standard form, with itis ability to predict buckling loads of the silo segment stiffened with thin-walled columns. The characteristic curves of the perfect and imperfect structure are traced and analysed.

Four primary vibration and buckling modes of the silo segment without initial imperfections are shown in Figure 5. It may be observed that the buckling modes shapes differ from the vibration modes completely. However, during structural loading process selected vibration modes change their shapes. While the first and fourth vibration modes remain unaffected by the applied load, the second and third modes continuously alter their forms. In our case, at first, the half-wave along the silo height of the second vibration mode narrowed making it possible for two new smaller waves to occur, forming three semi-waves along the silo height. This shape resembled the second buckling mode. Similarly, the third vibration mode consisting of two semi-waves along the silo height changed its form in the same way, narrowing and allowing two new semi-waves to occur, creating a total of four semi-waves. This shape, in turn, resembled the first buckling mode. The changes in vibration modes during the loading course are presented in Figure 6. When the applied load exceeded the maximum limit point, it was impossible to follow the vibration modes in some cases of the investigated structures with initial geometrical imperfections.

The full equilibrium paths between the average vertical displacements of the upper edges of the columns and the applied compressive loads are presented in Figure 7. The diagrams include the silo parts with various imperfection shapes and amplitudes. Denotations of the abbreviations used in the figure description are following: B0—the perfect geometry, B1—the initial imperfections as the first buckling mode, B2—the initial imperfections as the second buckling mode, F1—the initial imperfections as the first vibration mode, F2—the initial imperfections as the second vibration mode. The terms ‘2 mm’, ‘5 mm’ and ‘20 mm’ denote imperfection amplitudes. The imperfect geometry with the greatest assumed imperfection magnitude reduces the limit load over three times. Unsurprisingly, the buckling modes set as initial imperfections were more disadvantageous than the vibration modes (about twice). Moreover, the first buckling mode influenced the limit load slightly more than the second one. Note that the first and second buckling loads are close, however their corresponding buckling modes diverge. On the contrary, the first vibration mode showed a much smaller impact on the structural load-carrying capacity (P_lim_) than the second one.

Figure 8 and Figure 9 show the vibration correlation technique results. In each case four primary squared natural frequencies were followed. The diagrams show various relationships between natural frequencies and the applied load. Depending on geometrical imperfection shapes and amplitudes, the first natural frequency reaching the zero value was different. In a geometrically perfect structure the second and third frequencies reached zero almost at the same time (Figure 8A). Their corresponding vibration modes at these points resembled the first and second buckling modes. The predicted buckling load (P_pred_) was equal to 1383 kN, whereas the calculated buckling load was equal to 1360 kN. The relative difference was 1.7%. The limit load was equal to 1734 kN, i.e., 1.275 times greater than the buckling load. No initial imperfections introduced to the numerical model exceeded the outcome. Moreover, the first and fourth modes were straight, while the second and third ones (exhibiting changes during the loading course) non-linear. It should be noted that predicted buckling load is defined as applied load value either when natural frequency equals zero or when it reaches minimum value (if it does not intersect with load axis).

The limit loads of structures with various initial imperfections are frequently similar or much smaller than the numerical value of the lowest buckling load. The buckling loads predicted by means of the VCT represent, in contrary to the perfect structure, the limit load. The VCT seems to indicate the minimum of the lowest buckling load and the limit load as the predicted buckling load (P_pred_), not just the first of the two. In the case of the investigated shell the buckling load was smaller than the limit load for the perfect structure and greater than the limit load for the structure with imperfections (Table 1). This relation is also illustrated in Figure 10.

Equation (1) states that natural frequencies should tend to zero when the applied compressive load becomes a subsequent buckling load. However, numerical computations discard this statement. The squared natural frequencies followed one of two ways: they dropped to the zero value or they decreased to the minimum value at some level, next increasing. In almost all VCT diagrams one of the primary natural frequencies fell to zero; only in the structure with the second buckling mode as initial imperfection of 2 mm amplitude (Figure 8E) three lowest frequencies reached the minimum values and then rose again. In some cases it was difficult to match frequencies with correct vibration modes when the applied load was nearly equal to the maximum load capacity, therefore the curves corresponding to squared natural frequencies had stopped before they reached the load axis level (Figure 8E and Figure 9C,E).

Moreover, it can be observed that the greater the imperfection amplitude the more curvilinear the first squared frequency relation. The relationship between squares of natural frequencies and the compressive load was linear when the applied load was low or medium. While the limit load tends to its limit value the curve shapes become non-linear in a higher extent. When the curves frequently reached zero values rapidly, the VCT effectiveness was reduced.

In most cases of the conducted analyses the first squared frequency to reach the zero value was the second or the third one in a row. The second and third vibration modes resembled the first and second buckling modes. While the corresponding buckling loads were very close, the curves representing the second and third squared natural frequencies often dropped to zero at the same or nearly the same time. In the structure with the second buckling mode as initial imperfection of 2 mm amplitude the first and second curves reached the minimum values and no natural frequency was equal to zero (Figure 8E). When the initial imperfection took the buckling mode shape (first or second) with the greatest assumed amplitude (20 mm), the first natural frequency curve reached zero (Figure 8D,G). In both cases the second curve reached the minimum value, next it increased. When one of the three lowest natural frequencies were equal to zero, the other ones reached their minimum values and rose again. The exception was the perfect silo segment (Figure 8A), where the first and fourth curves were unaffected by the loading process. It may be incorrect in the case of vibration mode shapes assumed as initial geometrical imperfections because full analyses could not be finished due to the problems with determining the correct numbering of vibration modes and, in consequence, natural frequencies.

The predicted by the VCT buckling or limit loads are shown in Table 1. Denotations are as follows: P_lim_ is the maximum load-carrying capacity (limit load) based on non-linear analysis, P_cr_ is the lowest buckling load obtained from the linear buckling analysis, P_pred_ is the predicted load received in the VCT and the difference is assessed between the P_pred_ and the minimum of P_lim_ and P_cr_. If the zero value was not achieved by any curve, the predicted buckling load was assumed equal to the load when the squared natural frequency reached its minimum value. It can be noted that the difference was less than 2.5% in a majority of structural analyses. Only ‘B1_20 mm’ and ‘B2_20 mm’ structures, where large imperfection amplitudes were introduced, showed lower effectiveness of the VCT. It should be also highlighted that the predicted maximum load in the structures, where the buckling modes were introduced as initial imperfections, was always greater than the P_lim_ or P_cr_, it leads to underestimation of structural load-carrying capacity.

Figure 11 and Figure 12 illustrate the relationship between 1 − (f_n_/f_n0_)^2^ and (1 − P/P_n_)^2^ investigated in each structure. A squared drop of the load-carrying capacity ξ^2^ was determined according to the modified VCT proposed by Arbelo et al. [13]. Next, Equation (2) was employed to assess the predicted buckling load. The results are collected in Table 2. The buckling loads are equal or close to the results of the standard VCT. The maximum relative differences of both methods are 1.7%.

## 4. Conclusions

The following conclusions stem from the performed numerical analysis only.
The vibration correlation technique seems to be very effective in estimating buckling or limit loads of silos strengthened with columns sparsely distributed along their circumference.The VCT allows to predict the silo segment buckling load for the perfect structure and the limit load in the case of the imperfect structure.The increase of geometrical imperfection magnitude reduces the VCT precision in the case of the structures with buckling loads as initial imperfections.The loading course in numerical simulations affects the shape of selected vibration modes.The relationship between squared natural frequencies and the applied load is linear in small load cases only. Considerable non-linearity occurs when the applied load becomes close to the minimum buckling load or the limit load.

The future research will be focused on sensitivity analysis application in the VCT in order to improve the estimation of real buckling loads by means of experimental tests. A number of experiments is planned to be conducted in laboratory conditions to compare numerical and experimental results.

## Figures and Tables

**Figure 1 materials-14-00567-f001:**
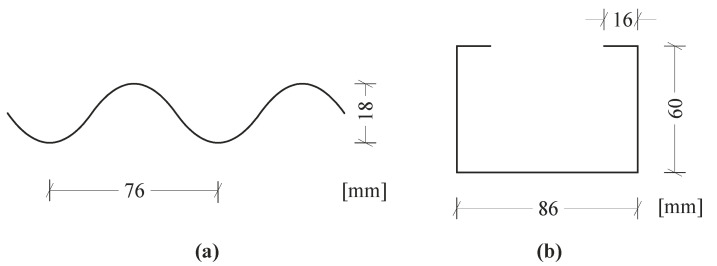
Dimensions of silo cross-sections: (**a**) corrugated sheet (silo wall); (**b**) cold-formed channel section (columns).

**Figure 2 materials-14-00567-f002:**
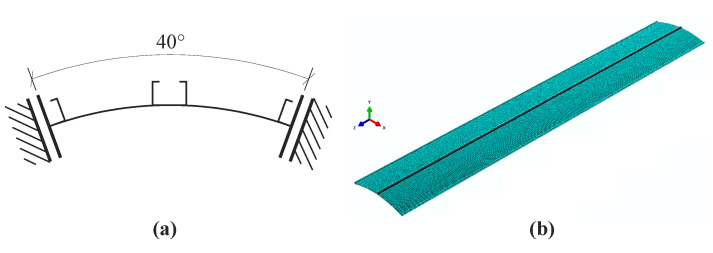
(**a**) Scheme of boundary conditions in simplified silo segment; (**b**) Visualisation of numerical model.

**Figure 3 materials-14-00567-f003:**
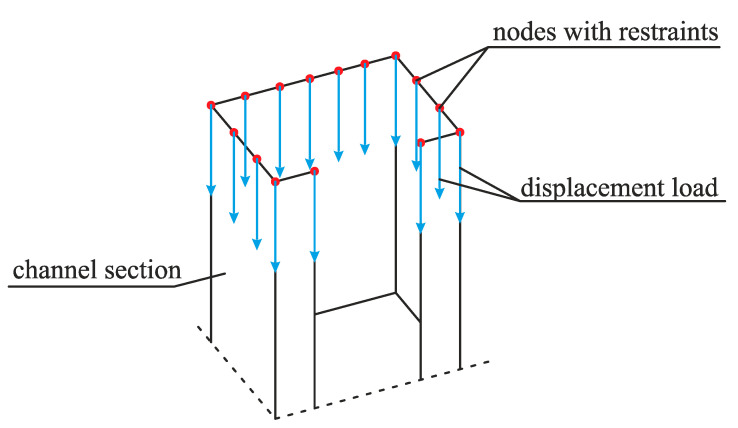
A pattern of displacement load subjected to the upper edges of silo columns.

**Figure 4 materials-14-00567-f004:**
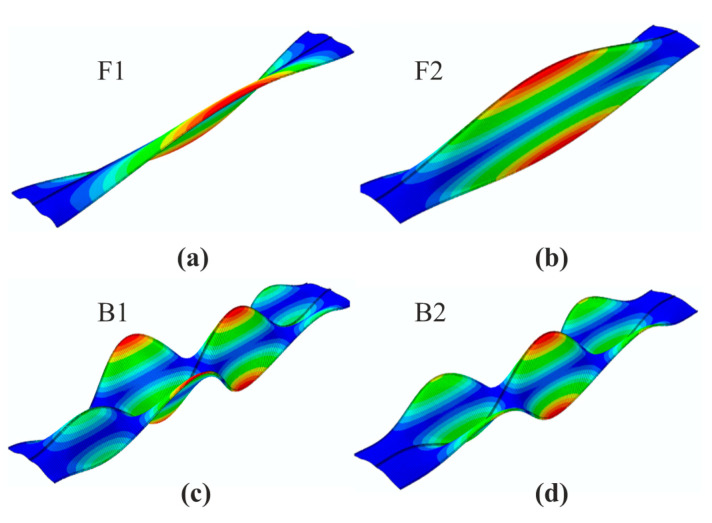
Initial geometrical imperfections: (**a**) first vibration mode; (**b**) second vibration mode; (**c**) first buckling mode; (**d**) second buckling mode.

**Figure 5 materials-14-00567-f005:**
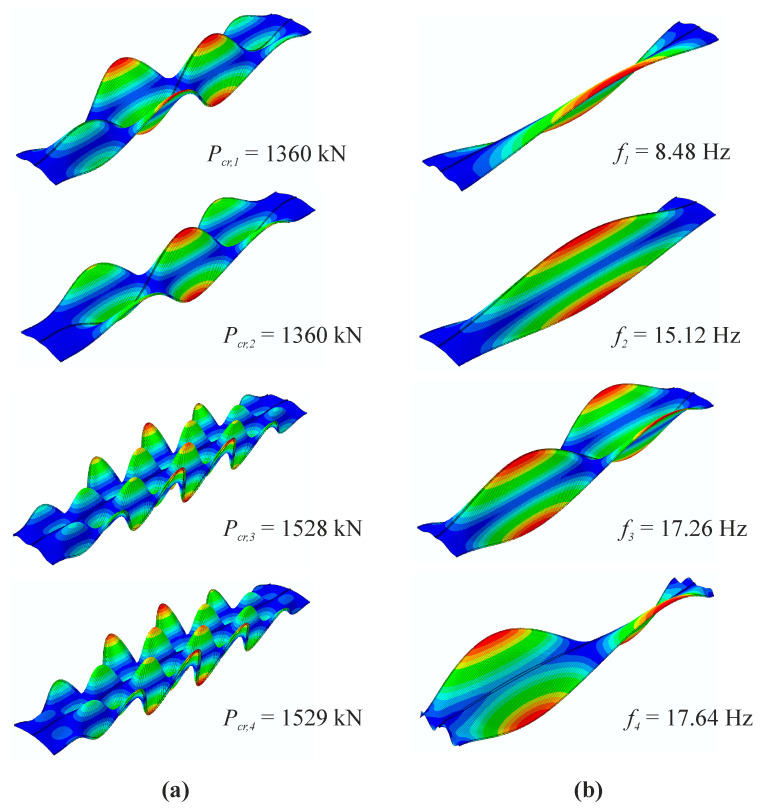
Primary one-to-four modes: (**a**) buckling; (**b**) vibration.

**Figure 6 materials-14-00567-f006:**
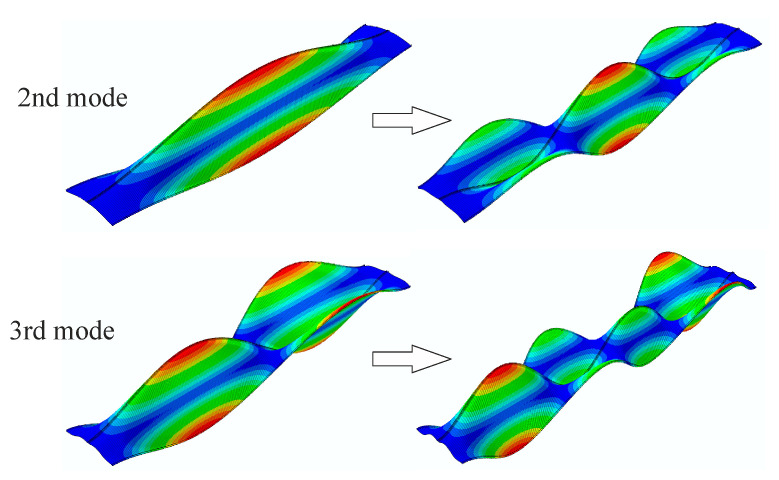
Changes of second and third vibration modes during loading course.

**Figure 7 materials-14-00567-f007:**
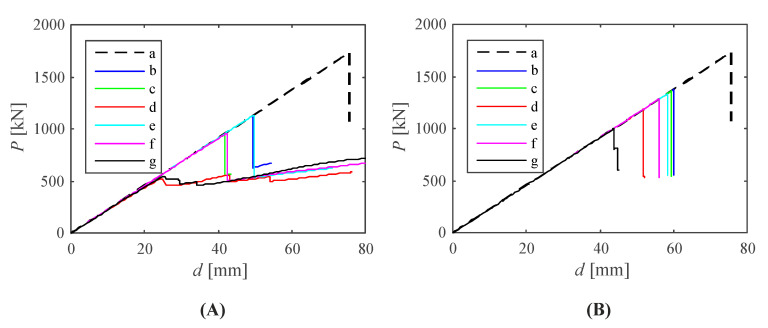
Evolution of total vertical reactions and average vertical displacement of columns top edges obtained from geometrically non-linear analyses of the silo segment with initial imperfections in the form of: (**A**) buckling modes: (a) B0, (b) B1_2 mm, (c) B1_5 mm, (d) B1_20 mm, (e) B2_2 mm, (f) B2_5 mm, (g) B2_20 mm; (**B**) vibration modes: (a) B0, (b) F1_2 mm, (c) F1_5 mm, (d) F1_20 mm, (e) F2_2 mm, (f) F2_5 mm, (g) F2_20 mm.

**Figure 8 materials-14-00567-f008:**
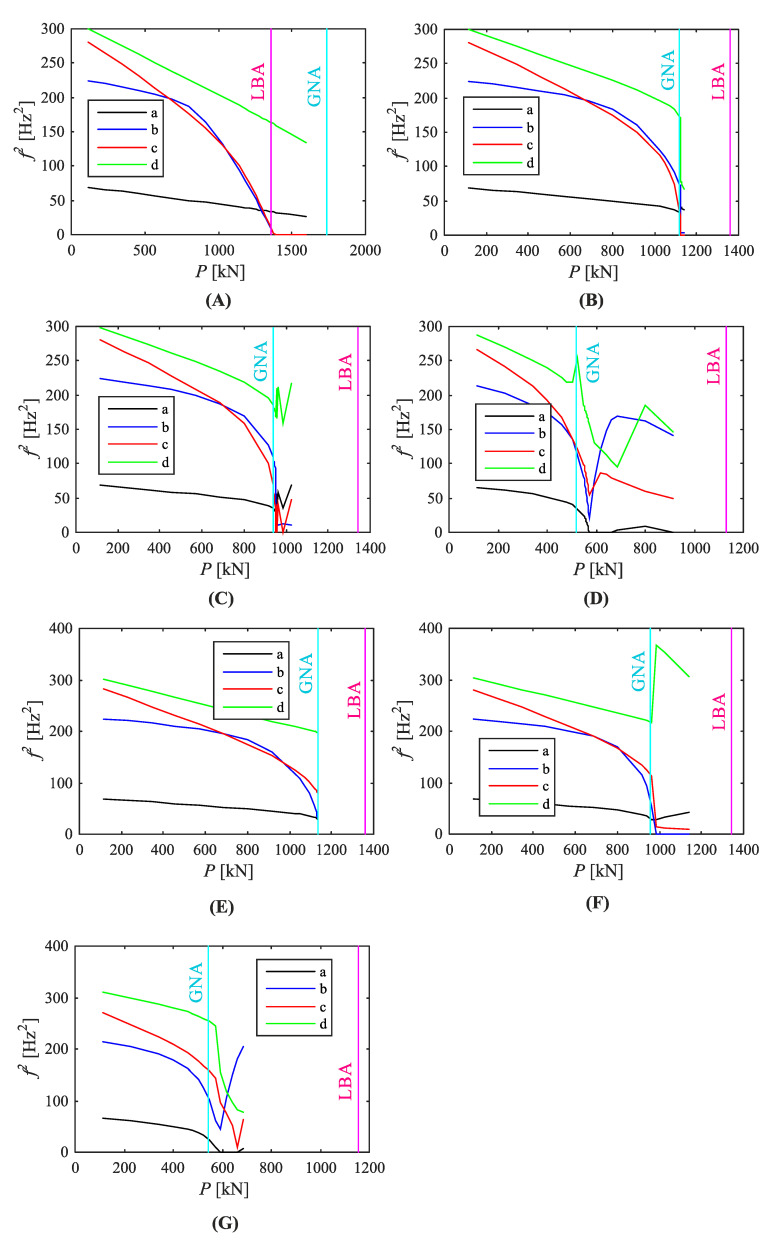
Relationship between applied load and squared natural frequencies ((a) first, (b) second, (c) third, (d) fourth) of the silo segment with initial geometrical imperfections: (**A**) none, (**B**) B1_2 mm, (**C**) B1_5 mm, (**D**) B1_20 mm, (**E**) B2_2 mm, (**F**) B2_5 mm, (**G**) B2_20 mm.

**Figure 9 materials-14-00567-f009:**
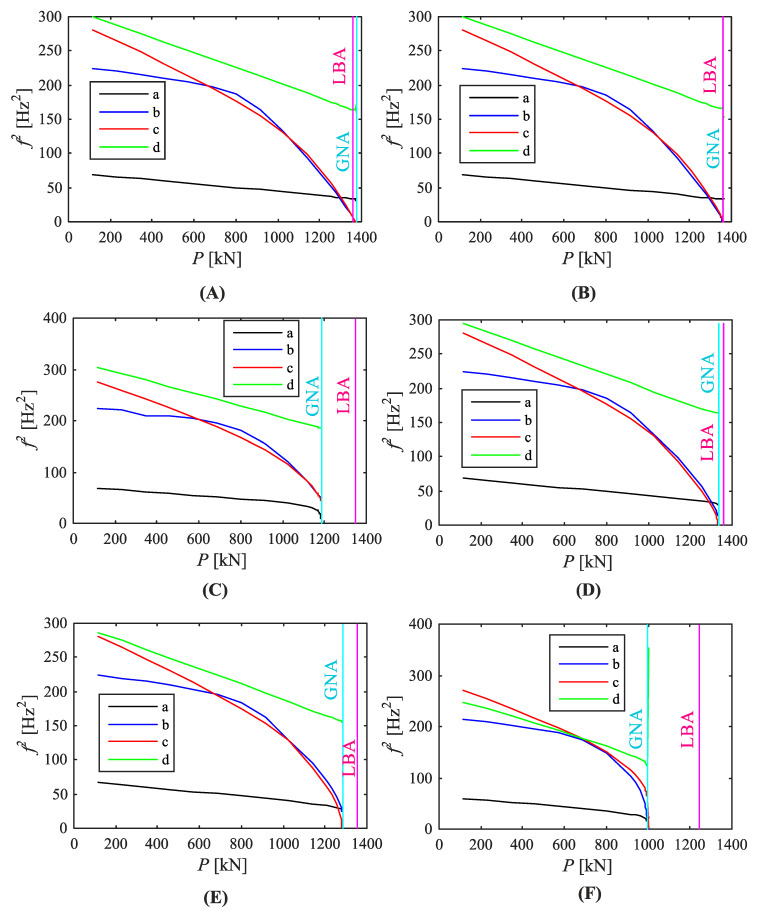
Relationship between applied load and squared natural frequencies ((a) first, (b) second, (c) third, (d) fourth) of the silo segment with initial geometrical imperfections: (**A**) F1_2 mm, (**B**) F1_5 mm, (**C**) F1_20 mm, (**D**) F2_2 mm, (**E**) F2_5 mm, (**F**) F2_20 mm.

**Figure 10 materials-14-00567-f010:**
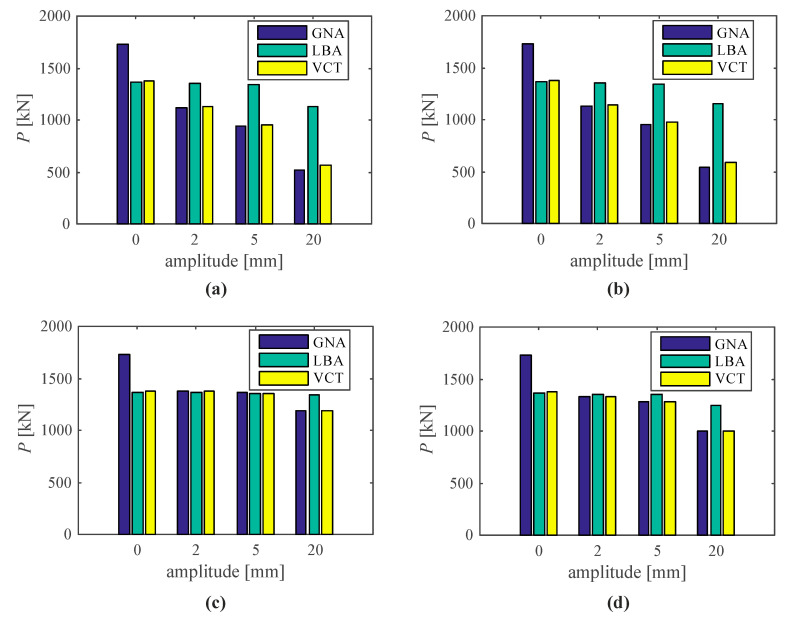
GNA, LBA and VCT loads versus imperfection amplitude of the silo segment with initial geometrical imperfections as modes: (**a**) first buckling (B1), (**b**) second buckling (B2), (**c**) first vibration (F1), (**d**) second vibration (F2).

**Figure 11 materials-14-00567-f011:**
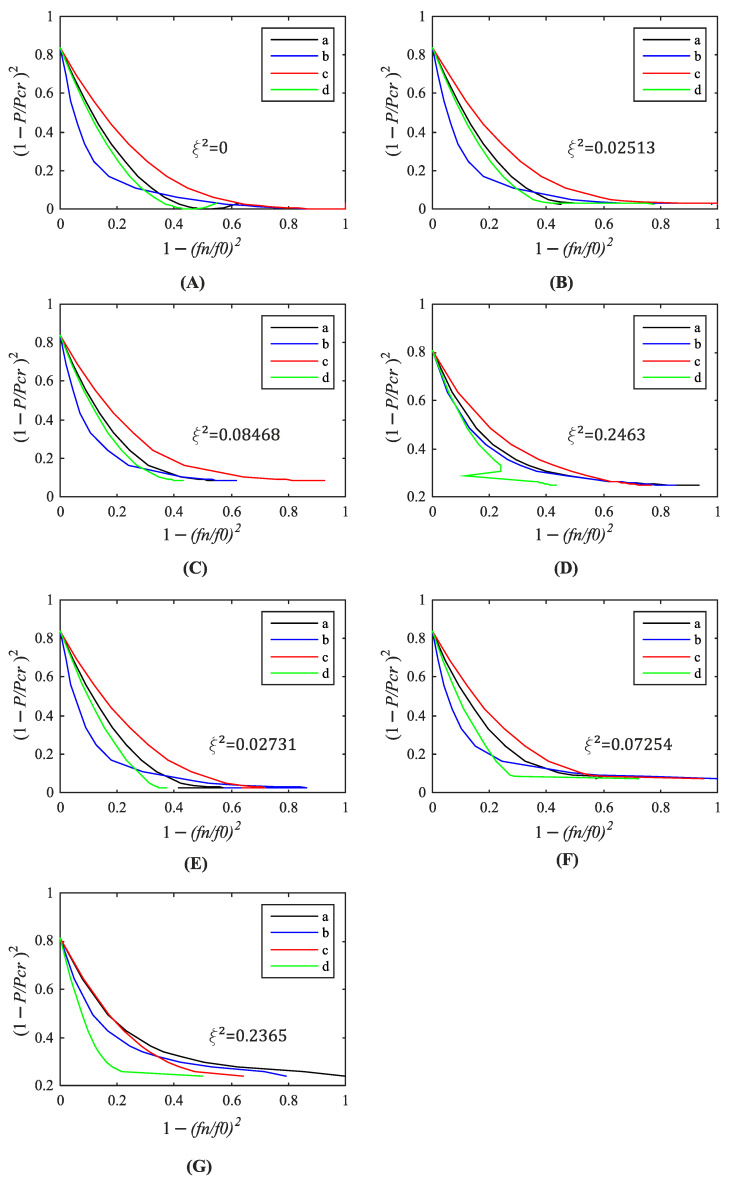
Relationship between relative load and natural frequencies ((a) first, (b) second, (c) third, (d) fourth) of the silo segment with initial geometrical imperfections: (**A**) none, (**B**) B1_2 mm, (**C**) B1_5 mm, (**D**) B1_20 mm, (**E**) B2_2 mm, (**F**) B2_5 mm, (**G**) B2_20 mm, according to the modified VCT by Arbelo.

**Figure 12 materials-14-00567-f012:**
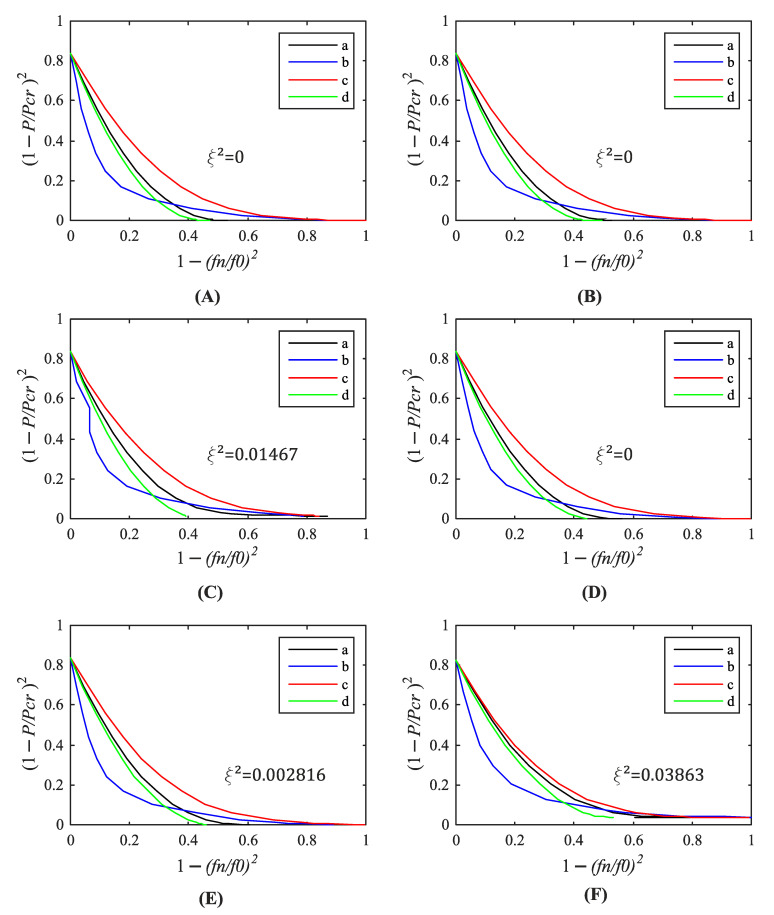
Relationship between relative load and natural frequencies ((a) first, (b) second, (c) third, (d) fourth) of the silo segment with initial geometrical imperfections: (**A**) F1_2 mm, (**B**) F1_5 mm, (**C**) F1_20 mm, (**D**) F2_2 mm, (**E**) F2_5 mm, (**F**) F2_20 mm, according to the modified VCT by Arbelo.

**Table 1 materials-14-00567-t001:** Summary of the GNA, LBA and VCT results.

The Numerical Model	P_lim_ [kN] (GNA)	P_cr_ [kN] (LBA)	P_pred_ [kN] (VCT)	The Difference [%]
B0	1734	1360	1383	1.7
B1_2 mm	1120	1357	1126	0.5
B1_5 mm	938	1340	952	1.5
B1_20 mm	517	1129	571	10.4
B2_2 mm	1134	1357	1142	0.7
B2_5 mm	958	1344	982	2.5
B2_20 mm	544	1156	594	9.2
F1_2 mm	1379	1360	1372	−0.9
F1_5 mm	1361	1359	1356	−0.2
F1_20 mm	1186	1346	1184	−0.2
F2_2 mm	1336	1359	1332	−0.3
F2_5 mm	1284	1353	1281	−0.2
F2_20 mm	997	1245	1000	0.3

**Table 2 materials-14-00567-t002:** Arbelo method results.

The Numerical Model	ξ^2^	P_cr_ [kN]	P_imp_ [kN]	The Difference [%]
B0	0	1360	1360	0
B1_2mm	0.02513	1357	1142	2.0
B1_5mm	0.08468	1340	950	1.3
B1_20mm	0.2463	1129	569	10.0
B2_2 mm	0.02731	1357	1133	−0.1
B2_5 mm	0.07254	1344	982	2.5
B2_20 mm	0.2365	1156	594	9.2
F1_2 mm	0	1360	1360	0
F1_5 mm	0	1359	1359	0
F1_20 mm	0.01467	1346	1183	−0.3
F2_2 mm	0	1359	1359	1.7
F2_5 mm	0.002816	1353	1281	−0.2
F2_20 mm	0.03863	1245	1000	0.3

## Data Availability

Data is contained within the article.

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
