# Peer review of "Impact of Geometrical Imperfections on Estimation of Buckling and Limit Loads in a Silo Segment Using the Vibration Correlation Technique"

_materials, 2021, doi:10.3390/ma14030567_

Round 1
Reviewer 1 Report
The manuscript presents a numerical study on the effectiveness of the vibration correlation technique (VCT) to predict buckling of a silo segment.
The VCT approach is based on experimental data only and any other use of the VCT like in the present paper , where numerical data is used misses the essence of the VCT.
The authors should address this issue in detail.
Chapter 2, Theoretical background, should be removed as it deals with beams and the present manuscript does not deal with beams.
Graphs of the frequency vs. axial load , as proposed by Arbelo should be presented to understand the how the results in Table 2 were obtained.
It is not clear how the axial compression load was applied to the silo strip. Please add a drawing to understand the topic.
Author Response
Dear Sir,
We express our deep gratitude to the Reviewer for all valuable questions and remarks, which allowed us to significantly upgrade our paper. We hope our corrections have made it more accessible.
Please, find below our detailed answers to the questions raised in the review.
The VCT approach is based on experimental data only and any other use of the VCT like in the present paper, where numerical data is used misses the essence of the VCT.
The authors should address this issue in detail.
We agree that it would be better to perform experimental analyses as well as numerical ones. However, our aim was to investigate a structure with realistic dimensions. It is difficult to perform such an experiment in 1 to 1 scale due to large silo dimensions, therefore we decided to limit investigations only to the numerical analyses. The VCT method was verified in a series of tests of small scale for cylindrical shells that were compared with results of numerical analyses in published papers (see Ref. 3, 12, 13) with a good result. We based on their conclusions. Our future work will focus on smaller structures which can be tested in laboratory conditions.
Chapter 2, Theoretical background, should be removed as it deals with beams and the present manuscript does not deal with beams.
We are grateful for the remark. We removed this chapter according to Reviewer suggestion.
Graphs of the frequency vs. axial load , as proposed by Arbelo should be presented to understand the how the results in Table 2 were obtained.
We added these graphs to the manuscript (Figures 11 and 12). We also put the squared drop of the load-carrying capacity in the figures. Now it should be more understandable for readers.
It is not clear how the axial compression load was applied to the silo strip. Please add a drawing to understand the topic.
The restrained nodes in the upper edges of the columns were forced to translate vertically along the column longitudinal axes. We added a Figure 3 illustrating this issue.
Please, take our words of gratitude for preparing the review with an intention to improve our paper.
Yours sincerely
Iwicki Piotr
Ł. Żmuda-Trzebiatowski
Reviewer 2 Report
accpet
Author Response
Dear Sir,
Thank you for the positive review of our paper. We made some minor linguistic corrections according to the opinion presented in the review. We hope our corrections have made it more accessible.
Changes to the article are marked in the new version in yellow.
Please, take our words of gratitude for preparing the review with an intention to improve our paper.
Yours sincerely
Piotr Iwicki
Ł. Żmuda-Trzebiatowski
Reviewer 3 Report
The authors have presented a method to determine buckling characteristics of curved shell structural elements via vibrational correlation analysis. A finite element model is described which is used to predict buckling modes and vibrational modes via linear eigenvalue analysis; a nonlinear analysis including imperfection modes based on these buckling / vibrational modes is conducted in order to obtain ultimate loads. The effect of various initial imperfection modes and amplitudes are assessed via a combined quasi-dynamic analysis whereby the model is pre-loaded and the vibrational and frequencies extracted. Prior to full acceptance of the article, the authors are advised to address the following issues.
- Line 46: consider rephrasing "draw a straight line"...perhaps "in order for there to be a linear relationship between the squared natural frequency and the buckling load".
- Line 100: for clarity, better to define that the subscripts denote the ordinates with respect to which variables are differentiated.
- Line 108: include a space between Given and Euler.
- Line 150: better to write "eigenvalues" rather than "coefficients", assuming that is the output here.
- Line 154: what load / displacement state is imposed in the static, general analysis? Is there any reason this was chosen over a Riks analysis?
- The presentation of the work would be served by defining the limit load obtained from the nonlinear analysis as N_u,NL, or similar, in order to reduce ambiguity.
- Line 161: how many pre-load steps were imposed when tracing out the frequency-load curves?
- Line 202: could make reference here to the fact that the critical loads pertaining to the first and second buckling modes are almost identical.
- Line 225: ...as would be expected when including initial imperfections, surely?
- Table 1 - see comment 6 about defining symbols to refer to the various ultimate / buckling loads.
- Line 257: "buckling modes".
Author Response
Dear Sir,
We express our deep gratitude to the Reviewer for all valuable questions and remarks, which allowed us to significantly upgrade our paper. We hope our corrections have made it more accessible.
Please, find below our detailed answers to the questions in the review.
- Line 46: consider rephrasing "draw a straight line"...perhaps "in order for there to be a linear relationship between the squared natural frequency and the buckling load".
We changed the sentence according to your suggestion.
- Line 100: for clarity, better to define that the subscripts denote the ordinates with respect to which variables are differentiated.
- Line 108: include a space between Given and Euler.
We appreciate these comments. However, we removed the chapter these lines refer to due to one of the reviewers’ suggestion.
- Line 150: better to write "eigenvalues" rather than "coefficients", assuming that is the output here.
Thank you for the comment. We agree that “eigenvalues” fits better in the sentence than “coefficients”, therefore we changed it.
- Line 154: what load / displacement state is imposed in the static, general analysis? Is there any reason this was chosen over a Riks analysis?
We imposed the displacement load in the static, general analysis. The restrained nodes in the upper edges of the columns were forced to translate vertically along the column longitudinal axes. We added a Figure 3 illustrating this issue.
There was no need to use the Riks method because the static, general non-linear analysis allowed to achieve our aim which was the full equilibrium path by controlling the displacement. Moreover, in the Abaqus software the dynamic eingenvalue analysis in the next step was easier to perform by using the static, general method.
- The presentation of the work would be served by defining the limit load obtained from the nonlinear analysis as N_u,NL, or similar, in order to reduce ambiguity.
We are grateful for this remark. We consented to your suggestion and implemented a denotation Plim defined by the limit load obtained in the non-linear analyses into the text.
- Line 161: how many pre-load steps were imposed when tracing out the frequency-load curves?
A number of pre-load steps was different for numerical models with various initial imperfections. The range was from 13 to 25, mostly 19 to 21.
- Line 202: could make reference here to the fact that the critical loads pertaining to the first and second buckling modes are almost identical.
Thank you for the suggestion. We added a new sentence (line 192) referring to the above-mentioned fact.
- Line 225: ...as would be expected when including initial imperfections, surely?
This comment made us reconsider the statement. It is very often true, however in some cases the limit load increases due to initial imperfections. We changed the sentence (line 223).
- Table 1 - see comment 6 about defining symbols to refer to the various ultimate / buckling loads.
We changed headings in the table according to your suggestion.
- Line 257: "buckling modes".
Thank you for catching this mistake. It was corrected.
Please, take our words of gratitude for preparing the review with an intention to improve our paper.
Yours sincerely
P. Iwicki
Ł. Żmuda-Trzebiatowski
Reviewer 4 Report
From a global point of view, the article is well structured and presented. The topic is interesting. The reviewer has some issues that could be clarified:
- The VCT technique has been shown to be efficient for calculating the limit load or buckling load due to imperfections for silos strengthened with columns sparsely distributed along their circumference. The authors should explain why the method has only been applied to these types of silos.
- When the geometric imperfection increases, the precision of VCT decreases. A more detailed and depth analysis of the causes of this fact is advisable.
- Authors are recommended to review the conclusions, showing through bullets more concise and clear conclusions that emerge from the research.
Round 2
Reviewer 1 Report
The VCT approach is an experimental procedure. Any other use of the VCT , like it is presented in the manuscript misses the whole advantage of the method, namely accurate nondestructive prediction of the buckling load of a thin walled structure.
The authors wrote in the introduction: "However, the VCT does not take any imperfections into account, therefore the method is generally applicable for simple structures. If the real imperfections are measured and introduced into the model, the VCT will work well for more complex structures"
This statement is no correct. See the literature. Please rephrase this paragraph.
The conclusions written are not true, and even wrong:
"The vibration correlation technique seems very effective in estimating buckling or limit loads of silos strengthened with columns sparsely distributed along their circumference".
This statement is based on a numerical analysis and it should be mentioned.
The method cannot be used for structures with large geometrical imperfections, which significantly affect the predicted loads.
This statement is not true. Check the literature. Rephrase it.
Great attention should be paid to vibration modes due to their variations during the loading course.
Correct statement.
Unfortunately, the relationship between squared natural frequencies and the applied load is linear only when the load is small. Considerable deviations from linearity occur when the applied load is almost equal to the minimum buckling load or maximum load-carrying capacity. Therefore experimental tests are much less useful in predicting buckling loads. However, they can be still applied to validate numerical models of existing structures, especially boundary conditions and global stiffness of investigated structures.
This conclusion is wrong and untrue. Remove it , or provide convincing evidence. The work presented in the manuscript is a numerical one, and you cannot state anything about experimental results.
Please refrain from reaching conclusions for the experimental evaluation and refer only to the numerical results, you had calculated.
Author Response
Dear Sir,
We express our deep gratitude to the Reviewer for all valuable questions and remarks, which allowed us to significantly upgrade our paper. We hope our corrections have made it more accessible. We marked them in the manuscript in green.
Please, find below our detailed answers to the questions raised in the review.
The VCT approach is an experimental procedure. Any other use of the VCT , like it is presented in the manuscript misses the whole advantage of the method, namely accurate nondestructive prediction of the buckling load of a thin walled structure.
We agree that the VCT is mainly an experimental procedure and the use of numerical computations without experimental tests does not use the main advantage of this method. However, the experiments are based on previously performed numerical simulations. Natural frequencies of structures can be measured, but can also be calculated with computer programs. The essence of our work is to find the relationship between the squared natural frequencies and the applied load to the structure, and then to determine the buckling loads or limit loads by means of numerical calculations. Such analyses are worth performing before the experiments will be carried out. We find some experiments difficult to perform due to the size of the structures in contrast to the numerical simulations. A relevant comment was added to the manuscript (see lines 98-105).
The authors wrote in the introduction: "However, the VCT does not take any imperfections into account, therefore the method is generally applicable for simple structures. If the real imperfections are measured and introduced into the model, the VCT will work well for more complex structures"
This statement is no correct. See the literature. Please rephrase this paragraph.
We are grateful for the remark. We have changed the cited paragraph because we used unintentionally a shorthand. We have meant that the use of the VCT to determine real boundary conditions without scanning geometrical imperfections in complex structures will not be effective.
The conclusions written are not true, and even wrong:
"The vibration correlation technique seems very effective in estimating buckling or limit loads of silos strengthened with columns sparsely distributed along their circumference".
This statement is based on a numerical analysis and it should be mentioned.
We have added an appropriate comment at the beginning of the conclusions section in such a way that it is evident we based only on numerical analyses.
"The method cannot be used for structures with large geometrical imperfections, which significantly affect the predicted loads".
This statement is not true. Check the literature. Rephrase it.
This sentence is indeed formulated incorrectly. We meant that large geometrical imperfections cause a bigger difference between the buckling or limit load calculated numerically and the predicted buckling load by means of the vibration correlation technique. We have changed this sentence.
"Great attention should be paid to vibration modes due to their variations during the loading course".
Correct statement.
We have changed this statement.
"Unfortunately, the relationship between squared natural frequencies and the applied load is linear only when the load is small. Considerable deviations from linearity occur when the applied load is almost equal to the minimum buckling load or maximum load-carrying capacity. Therefore experimental tests are much less useful in predicting buckling loads. However, they can be still applied to validate numerical models of existing structures, especially boundary conditions and global stiffness of investigated structures".
This conclusion is wrong and untrue. Remove it , or provide convincing evidence. The work presented in the manuscript is a numerical one, and you cannot state anything about experimental results.
We agree that we should not mention about experimental tests which were not performed. We have removed two last sentences concerning experiments. The first two sentences base on the numerical analyses and the VCT diagrams presented in the manuscript.
Please, take our words of gratitude for preparing the review with an intention to improve our paper.
Yours sincerely
P. Iwicki
Ł. Żmuda-Trzebiatowski
Reviewer 4 Report
The suggested issues have been adequately clarified.